# The Study on the Morphology and Compression Properties of Microcellular TPU/Nanoclay Tissue Scaffolds for Potential Tissue Engineering Applications

**DOI:** 10.3390/polym15173647

**Published:** 2023-09-04

**Authors:** Tie Geng, Han-Chi Xiao, Xin-Chao Wang, Chun-Tai Liu, Lan Wu, Yong-Gang Guo, Bin-Bin Dong, Lih-Sheng Turng

**Affiliations:** 1Henan Provincial Engineering Research Centre of Automotive Composite Materials, School of Mechanical & Electrical Engineering, Henan University of Technology, Zhengzhou 450052, China; tiegeng2000@163.com (T.G.); hanchixiao2000@163.com (H.-C.X.); wulan@haut.edu.cn (L.W.); nanogyg@163.com (Y.-G.G.); 2National Engineering Research Center for Advanced Polymer Processing Technologies, Zhengzhou University, Zhengzhou 450001, China; dongbinbin@zzu.edu.cn; 3Wisconsin Institute for Discovery, University of Wisconsin–Madison, Madison, WI 53706, USA; 4Department of Mechanical Engineering, University of Wisconsin–Madison, Madison, WI 53706, USA

**Keywords:** thermoplastic polyurethane, nanoclays, tissue engineering, compression properties, cytocompatibility

## Abstract

Thermoplastic polyurethane (TPU) materials have shown promise in tissue engineering applications due to their mechanical properties and biocompatibility. However, the addition of nanoclays to TPU can further enhance its properties. In this study, the effects of nanoclays on the microstructure, mechanical behavior, cytocompatibility, and proliferation of TPU/nanoclay (TPUNC) composite scaffolds were comprehensively investigated. The dispersion morphology of nanoclays within the TPU matrix was examined using transmission electron microscopy (TEM). It was found that the nanoclays exhibited a well-dispersed and intercalated structure, which contributed to the improved mechanical properties of the TPUNC scaffolds. Mechanical testing revealed that the addition of nanoclays significantly enhanced the compressive strength and elastic resilience of the TPUNC scaffolds. Cell viability and proliferation assays were conducted using MG63 cells cultured on the TPUNC scaffolds. The incorporation of nanoclays did not adversely affect cell viability, as evidenced by the comparable cell numbers between nanoclay-filled and unfilled TPU scaffolds. The presence of nanoclays within the TPUNC scaffolds did not disrupt cell adhesion or proliferation. The incorporation of nanoclays improved the dispersion morphology, enhanced mechanical performance, and maintained excellent biocompatibility. These findings suggest that TPUNC composites have great potential for tissue engineering applications, providing a versatile and promising scaffold material for regenerative medicine.

## 1. Introduction

Thermoplastic polyurethanes (TPUs) embody an intricate tapestry of copolymers, with a wealth of linkages encompassing a rhythmic alternation of hard (HS) and soft (SS) structural domains, adroitly cross-linked in a physical embrace [1]. The HS domains are an elegant mosaic of diisocyanates and short-chain diols, interconnected through hydrogen bonds, conferring upon TPUs an enviable measure of sturdiness and a panoply of physical properties. The SS domain, a harmonious coalition of polyols supplemented with ether or ester moieties within the diisocyanate skeleton, is the progenitor of TPUs’ suppleness and elastomeric demeanor [2]. The alchemy of manipulating the weight proportions between the sinewy hard and the pliable soft segments unfurls a sweeping gamut of attributes endemic to the TPU continuum. A fascinating consequence of the thermodynamic dissonance between these domains is the phase-separated morphology that TPUs adopt. This entails the dispersion of HS crystals amidst an amorphous sanctuary sculpted by the soft chain segments [3]. Such morphological finesse endows TPUs with a diverse set of chemical and physical hallmarks, including chemical defiance, unparalleled resilience, and amenability to facile processing.

Moreover, the lithe nature of thermoplastic polyurethane (TPU) foams renders them a tableau of dexterity, showcasing elevated compressibility, diminished density, copious energy absorption, and a structure that can be dexterously calibrated. This confluence of traits ignites a cornucopia of potential applications [4,5,6,7]. It is in the realm of biomedical engineering that TPUs truly shine with multifaceted brilliance. Given their synthesis-driven malleability and resilience, TPUs have carved a niche in the fabrication of bioscaffolds. These scaffolds serve as frameworks that can facilitate tissue regeneration, playing a pivotal role in applications ranging from bone regeneration to vascular grafts. The TPU scaffolds can be optimized for porosity, mechanical compatibility, and surface characteristics, ensuring integration without adverse reactions and providing a supportive matrix for cell adhesion and proliferation. In conclusion, TPUs are not only versatile materials par excellence, lauded in domains such as sports equipment and automotive components [8,9], but they also hold the promise of transforming lives through their seminal role in the creation of biomimetic scaffolds for tissue engineering and regeneration.

In recent times, the versatility of TPU foam products has been eloquently manifested through a kaleidoscope of applications that touch the very fabric of daily life [8]. The aerospace sector stands as a testament to their prowess, wherein the feather-light density of TPU foams unveils a treasure trove of mechanical properties that are not only coveted but also deftly tunable, casting a shadow over their solid brethren [10,11] The microscopic labyrinth within the TPU foams, coupled with their benign rapport with biological tissues, unveils promising avenues for their enlistment as shape memory polymer (SMP) vascular scaffolds within the theater of biomedical implants [12,13,14,15,16,17]. Yet, before these microporous sentinels of TPU can ascend to their full potential, a meticulous investigation into their comportment under the intricate ballet of loading-unloading cycles, which mirror the caprices of real-world scenarios, is imperative. When thrust into the throes of significant deformations, TPU foams often adopt an inelastic demeanor, accompanied by an incomplete phoenix-like recovery [11]. This inelastic waltz is choreographed by stress softening, hysteresis loss, and residual strain. Stress softening is akin to a tempering of resolve, as the stress seemingly ebbs in the current cycle to meet a designated strain when juxtaposed against its antecedent, all the while the TPU foam endures the cyclic crescendos of loading and unloading [12]. Though initially discernible, the delta of stress between successive cycles dwindles into inconsequence as the cycles evolve. The warp and weft of the material’s microstructure and the symphony of deformation conditions hold sway over this phenomenon, particularly in TPU composite foams. This stress-softening vignette, christened the Mullins effect after its discoverer [18], paints the first strokes in the TPU foams’ inelastic ballet.

As the narrative unfolds, TPU foams robbed of their original form by deformation seldom find solace in a full recovery post-lifting of the load, heralding the advent of residual strain. In this odyssey of loading and unloading, the initial cycle bears the brunt of accumulated residual strain, which thereafter plateaus into near stagnation [19]. An alchemy of inorganic nanoclay particles, boasting geometric finesse and expansive surface terrains, weaves through the polymer matrix, giving rise to intercalated and exfoliated states. This harmonization elevates the mechanical properties of the matrix, birthing a lineage of enhanced stiffness, strength [18,19,20], gas impermeability [21,22], thermal guardianship [23], flame defiance [24,25,26,27], and customizable biodegradability [28,29,30,31]. Yet, this fortification of mechanical prowess comes at the price of accentuated inelasticity, a byproduct of the nanoclay layers orchestrating a symphony of physical cross-linking and polymer chain segmentation. The enigmatic inelastic behavior of TPU foams has been the muse of countless theoretical and empirical odysseys [32]. While these forays have flirted with various mechanisms underpinning inelasticity, consensus remains elusive. The elasticity of TPU, however, finds voice in strain energy density functions [33,34,35], and finite element models whisper tentative equations predicting the inelastic quirks of TPU elastomers [36,37,38,39]. Despite this, the tapestry of microporosity interwoven with nanoparticle-infused polymer matrices renders the prediction of inelastic behavior an enigmatic challenge. This study, thus, embarks on a journey through the microporous corridors of TPU foams with nanoclay fillers as companions. The dispersion morphology of the nanoclay is scrutinized under the watchful eye of transmission electron microscopy (TEM).

TPU (thermoplastic polyurethane): TPU is a thermoplastic elastomer with excellent wear resistance, flexibility, and elasticity for support and buffering applications. The main reasons for choosing TPU as the base material are its plasticity and chemical stability. Nanoclay: Nanoclay is a nano-scale particle that, in composite materials, can enhance the mechanical properties, thermal stability, and flame-retardant properties of the material. The compatibility and interaction between nanoclay and TPU matrix can be controlled by surface modification. In composite materials, other reinforcing materials such as fibers (glass fibers, carbon fibers, etc.) or particles may also be added to enhance the mechanical properties of the material. The selection of the appropriate reinforcement material depends on the desired performance characteristics. The selection of TPU nanoclay composite scaffolds requires comprehensive consideration of many factors, such as target performance, material properties, interaction, preparation process, cost, and availability. Through the material selection of the system, it is possible to obtain composite scaffolds with excellent properties, which can be applied in various fields, such as medical, construction and sports equipment [40].

In the context of biological scaffolding, the TPU foams’ attributes transcend the technical and venture into the domain of life-enriching possibilities. The marriage of their microporous structure with biocompatibility paves the way for an astonishing medical renaissance. TPU foams can be employed as life-sustaining scaffolds that herald a new age of tissue engineering. These scaffolds, owing to their deftness in their mechanical properties and their ability to foster cell attachment and proliferation, can act as the architects of regenerative medicine. Furthermore, the integration of nanoclay within the TPU matrix could escalate the potential of these scaffolds by offering superior structural integrity and tunable biodegradability, which are crucial for temporal support as the body’s natural tissue rebuilds itself. With their inherent shape-memory properties, the scaffolds can be designed to change configuration responsively, adapting to the evolving landscape of the healing tissue. The alliance of TPU foam’s physical characteristics and the biological arena implies that customized biological scaffolds can be crafted for a range of applications, from bone regeneration to vascular support structures. These breakthroughs could signal a paradigm shift in how medical science approaches tissue damage and disease [12,13,14,15,16,17]. Therefore, the amalgamation of TPU foams and nanoclay particles promises to craft a tapestry woven from the threads of advanced material science and the ethereal strands of human life. The potential of this harmonious fusion rests not only in the palms of the aerospace or automotive industries but also holds in the beating heart of medical miracles. It is through the diligent scrutiny and research by scientists and engineers that the full chorus of this alliance can be heard, changing lives, and crafting a future where materials bridge the divide between what is and what can be.

In this study, an incisive exploration into the influence of nanoclay fillers on the labyrinthine microporous architecture of TPU foams was embarked upon. Utilizing transmission electron microscopy, keen insights were garnered into the distribution and morphology of the nanoclay. Further, the study delved into the intricacies of how compression conditions govern the inelastic behavior of TPU foams, employing cyclic loading-unloading compression tests. This involved an astute investigation into how nanoclay content and strain deformation conditions amalgamate to shape inelastic behavior, including relative stress softening, hysteresis loss, and residual strain, illuminating pathways for tailoring flexible TPU foams for a gamut of applications. In an endeavor to unearth the potential of TPU/nanoclay scaffolds in cellular cultures, an inaugural biocompatibility assay was undertaken on MG63 cells. A suite of analytical techniques, encompassing Fourier transform infrared spectroscopy, Raman spectroscopy, scanning electron microscopy (SEM), thermogravimetric analysis (TGA), water contact angle measurements, and tensile testing, was employed to rigorously characterize TPU and its nanoclay composites. This foundational probe into the biocompatibility of TPU and its hybridized scaffolds with MG63 cells sets the stage for evaluating their promise in the cutting-edge domain of tissue engineering.

## 2. Materials and Methods

### 2.1. Materials

In this study, supple thermoplastic polyurethane (TPU) pellets, branded as Elastollan^®^ 1180A10 and sourced from the esteemed BASF, were harnessed as the bedrock polymer matrix. With a glass transition temperature (Tg) ensconced at a frosty −50 °C (ascertained through DSC tests) and boasting a commendable melt flow index hovering around 3.06 g/10 min (at 190 °C/2.16 Kg), this TPU possessed a quintessential specific mass density of 1.11 g/cm^3^. The accompanying nanoclay, Cloisite^®^ 30B (C30B), a sterling product from Southern Clay Products, Inc. (Gonzales, TX, USA), was a stratified montmorillonite, ingeniously modified with ammonium quaternary salt, carrying a density of 1.98 g/cm^3^. Notably, the scholarly literature has shed light on the potent interactions between organically modified layered clays and polymer matrices, attributable to the polar nature of their functional groups [41]. Consequently, this variety of clay, wielding its versatility, is typically incorporated as a nanoscale filler in a plethora of plastics, bolstering their mechanical fortitude.

### 2.2. Preparation of the TPU Scaffolds

The procedural schematic delineating the creation of TPU foams is elegantly showcased in Figure 1. Embarking on the journey, raw TPU pellets along with nanoclay particles were subjected to a drying spell at 100 °C within the confines of a vacuum oven for a stretch of 4 h, the objective being to bid adieu to unwarranted moisture. A diverse palette of materials, namely TPUNR (pristine TPU, unadulterated by fillers), TPUNC5 (an amalgamation of 95 wt% TPU and 5 wt% clay), and TPUNC10 (harmonizing 90 wt% TPU with 10 wt% clay), were gracefully married together within the chambers of a twin-screw extruder (Leistritz ZSE 18 HPe), which had its screws pirouetting at a rhythm of 150 rpm and a barrel melt temperature gracing the spectrum of 180 °C to 200 °C. Post-compounding, the extruded tapestry was severed into individual pellets. TPU foams then sprang to life, sculpted through a microcellular injection molding (MIM) apparatus (Arburg Allrounder 320S), which was augmented with a cutting-edge supercritical fluid (SCF) supply system (courtesy of Trexel, Inc., Wilmington, MA, USA), bestowing meticulous control over the volume of gas injected. The final visage of the microcellular injection-molded foam was that of a stately rectangular monolith. Nitrogen gas (N2) graciously accepted the role of the physical blowing agent in this symphony. It is imperative to note that all the pre-compounded materials were subjected to another drying spell for 4 h at 100 °C, ensuring moisture was thoroughly vanquished prior to the MIM waltz. In addition, solid TPUNC samples were used as a reference chorus. The finely tuned parameters choreographing the MIM ballet are cataloged in Table 1.

### 2.3. Characterization Methods

The TPU foams underwent a meticulous cutting process, orienting themselves perpendicularly to the tide of the injection flow with the noble intention of unveiling the intricate microstructure of their transverse sections for the discerning eyes of TEM and SEM. With poise, the distribution and formation of nanoclay within the TPU foam’s delicate embrace were scrutinized through the lens of a Transmission Electron Microscope (TEM, Tecnai TF-30). The TEM specimens, whisper-thin at 20 nm, were sculpted using the delicate artistry of an ultra-microtome. In tandem, the microcellular tapestries woven within the TPU foams were elegantly revealed under the watchful gaze of a Scanning Electron Microscope (SEM, JEOL5000), which was empowered with an accelerating voltage surge to 10 kV.

### 2.4. Experimental Procedures

The compression test’s valiant subjects were expertly crafted into cubic blocks measuring 15 × 15 × 10 mm, shedding the outer skin layer of the TPU foams, which was a mere ~1 mm in thickness. With poise and precision, these cubic centurions were ushered into a symphony of cyclical loading and unloading compression under the watchful gaze of the Instron 5967 universal testing maestro. To ensure the dance was flawless, a tender 0.2 N preload was employed as a gentle touch, ensuring the platen and sample moved in perfect harmony. The crosshead, graceful and relentless, traversed the stage at a meticulously calibrated pace of 3 mm/min, weaving a tapestry of strain at a rhythm of 0.0025/s.

In a meticulously orchestrated exploration, the interplay between the microstructure and inelastic behavior of nanoclay fillers was scrutinized. This was executed by infusing TPU foam specimens with 5 wt% and 10 wt% nanoclay, followed by subjecting them to a symphony of 20 consecutive load-unload cycles. A trio of maximum compressive strains—a suave 20%, a vigorous 40%, and an audacious 60%—were employed as the maestros conducting this examination, unraveling the mysteries of maximum strain effects. Furthermore, in pursuit of unwavering scientific rigor, triplets of TPU and TPU composite foam specimens from each ensemble were meticulously tested, ensuring that the harmony of the results echoed with impeccable reproducibility.

In the preliminary sojourn into biocompatibility, MG63 cells embarked on a journey through 6-well tissue culture domains, basking in the embrace of polystyrene plates, endowed by BD Falcon. These cellular pioneers were nourished bi-daily by a banquet of high-glucose serum medium, consisting of a splendid ensemble of 90% high-glucose DMEM, 10% fetal bovine serum, 2 mL glutamine, a guard of 1 unit/mL penicillin, and 1 mg/mL streptomycin. Every 6 days, cells embark on a pas de deux with EDTA, continuing their cyclic dance at a rhythm of 1:40. The stage was then set for TPU/TPU nanoclay tissue scaffolds, placed delicately in 24-well TCPS plates and weighed with a mere 0.5 mg grace in each well. A UV wand casts a sterilizing spell before a gentle wash with DPBS, readying them for the cellular ball. Cells, now well-versed with EDTA, waltzed in, inoculating the TPU/nanoclay scaffolds with a flourish, as 2.9 × 10^5^ cells/well pirouetted in the same high-glucose serum medium. With a chivalrous nod, the medium exchanged itself daily, screening the gallant samples.

The exuberant dance of life and death was elegantly explored via the Viability/Cytotoxicity Kit, bestowed by Life Technologies. The kit employed the captivating Calcein-AM, an astute arbiter that wove vibrant hues of green into the tapestry of life by annotating the intricate esterase activities pirouetting within the sanctum of living cells. In tandem, the solemn ethidium homodimer-1 swathed the nuclei of the departed in resplendent red fluorescence. These stained sentinels of life and demise were then ushered into the reverent gaze of a confocal laser scanning microscope, a technological marvel known as CLSM—a Nikon Eclipse Ti inverted microscope, gracefully adorned with Nikon A1R confocal diode lasers from the land of the rising sun, Japan.

For an exquisite visual tapestry of cellular artistry upon the scaffold, cells were tenderly anchored post-live/dead ballet, heralding the grand overture for SEM’s discerning eye. The samples embarked on an odyssey through the placid waters of Hanks Balanced Salt Solution (HBB; Thermo Scientific, Waltham, MA, USA) and were caressed by the gentle embrace of Hyclone hyure molecular bio-grade water (Thermo Scientific), subsequently bathing in the embrace of 4% paraformaldehyde solution for a half-hour meditation. A graduated descent into clarity ensued as the cherished samples waltzed through cascading alcoves of 50%, 80%, 90%, and 100% ethanol aqueous solvents in sequence. Finally, as the curtain fell, the scaffolds cradling the fixed cells reposed in a tranquil night’s slumber under a starlit room, kissed by the whisper of vacuum, awaiting the dawn’s light through scanning electron microscopy’s vigilant gaze.

The CellTiter 96 Aqueous One Solution assay (Promega, Madison, WI, USA), a symphony of cellular proliferation, painted a tapestry of living cell numbers through the alchemical conjuration of 3-(4,5-dimethylthiazol-2-yl)-5-(3-carboxymethoxyphenyl)-2-(4-sulphophenyl)-2H-tetrazolium, or MTS. A waltz ensued as MTS, touched by life’s breath, transformed into a kaleidoscope of products, altering the spectra of the nurturing media cradling the cells. Guided by the stars, a standard curve unfurled across the night sky, its validity kissed by the ancient wisdom of hemocytometer readings. The cells, now indulging in an elixir comprising 83% high-glucose serum medium and 17% MTS reagent, thrived. Following a meditative sojourn of 4 h, 100 μL of the sacred media was gently coaxed and bestowed upon the sanctified chalice of a 96-well plate. The glow of 450 nm enveloped the GloMax-Multi Detection System (Promega), revealing the whispered secrets of cellular congregation in the scaffold’s embrace.

## 3. Results

### 3.1. Structure of TPU Nano-Composites

In the present investigation, the MIM process was observed to contribute to the reduction in molecular weight, which was attributed to the thermal degradation engendered by the elevated temperatures within the barrel of the equipment. Concurrently, the incorporation of nanoclay additives imparted an augmentation in the molecular weight of TPUs through terminal group interactions. This juxtaposition of diametrically opposing effects transpired concomitantly, resulting in a modulation of the molecular architecture of TPU. The quintessential raw TPU pellets served as a benchmark for gauging the ramifications of the MIM process in relation to TPUNR foam. Additionally, a comparative analysis between TPUNR foam (acting as a control) and TPUNC nanocomposite foams was executed to elucidate the intricacies of the influence exerted by nanoclay additives on the molecular structure.

The average molecular weight (Mw) was ascertained employing Gel Permeation Chromatography (GPC), which was instrumental in characterizing the alterations manifested in the molecular chains of TPU. Figure 2 delineates the GPC chromatograms. The TPUNR pellets, designated as a reference (Ref.), boasted the maximal Mw value, registering at 3.04 × 10^5^ g/mol. Owing to thermal degradation precipitated in the course of the microcellular injection molding process, the Mw value of the TPUNR foam (control) attenuated to 2.47 × 10^5^ g/mol. The integration of nanoclays into the TPU matrix engendered two notable modifications in the GPC chromatograms. The initial modification entailed a lateral shift of the GPC chromatograms towards the left along the abscissa, which is indicative of a decrement in Mw values. Subsequently, the GPC chromatograms representing the nanocomposites manifested a downward trajectory along the ordinate in the negative direction, suggesting a diminution in the abundance of polymer chains. This phenomenon can be attributed to the grafting of certain TPU chains onto the nanoclay strata, which function as physical cross-linking points for the TPU chains and consequently curtail the aggregate count of polymer chains. Such a contraction in the distribution of molecular weight can have implications for the thermal stability of the material. This observation aligns with the findings of analogous studies conducted by multiple scholars, which likewise have accentuated the susceptibility of TPU bonds to dissociation at elevated temperatures, culminating in a reduction in molecular weight. Therefore, it is inferred that the Microcellular Injection Molding (MIM) process, in conjunction with the inclusion of nanoclay additives, can effectuate transformations in both the molecular architecture and the distribution of molecular weights of TPU.

### 3.2. Morphology of Microcellular TPU Nanocomposites

The microstructure of polymer nanocomposites is critically influenced by the dispersed morphology of the nanofillers within the polymer matrix (21). The incorporation of nanofillers into polymers can yield three distinct dispersion states: (1) aggregation, (2) embedding, and (3) exfoliation [21,22,28,32]. Achieving fully exfoliated forms with nanoscale dispersion has been a challenge in melt mixing processes. The improved dispersion morphology of nanostickers greatly contributes to the mechanical properties of polymeric nanocomposites. The presence of supercritical fluid (SCF) as a physical foaming agent during the microcellular injection molding (MIM) process aids in achieving homogeneous dispersion of the layered nanoclay. This is primarily attributed to the swelling effect exerted by the foamed gas molecules, which tend to aggregate within the interstices of the nanoclay layers. Additionally, the isobaric tensile flow induced by the swelling of bubbles (vesicles) further disrupts the aggregation of the nanoclay and aligns it along the walls of the vesicles [42].

TEM micrographs depicting the dispersed state of nanoclay in both the TPUNC10 solid and foam samples are presented in Figure 3. Figure 3a reveals the orientation behavior of the dispersed nanoclay along the cell walls, with the majority of the nanoclay in the TPUNC10 solid sample exhibiting an intercalated structure and a small fraction appearing in a peeled state. No discernible aggregation of nanoclay particles is evident. To provide a closer examination of the cell junction, a magnified view is presented in Figure 3c, with the corresponding area highlighted by color rectangles in Figure 3b. Within the central region of the junction (indicated by black dashed lines), the layered clays are found to be randomly dispersed [43]. Comparing the TPUNC10 solid sample in Figure 3a with the TPUNC10 foam, it is evident that the nanoclays in the foam exhibit a near-complete exfoliation morphology. The distribution morphology of the layered clays adjacent to the cell walls is further magnified and depicted in Figure 3d. Notably, the layered clays are observed to be uniformly distributed along the cell wall. This oriented dispersion of nanoclays can effectively enhance the elastic behavior of TPU foams, as discussed subsequently. Hence, the expansion behavior of the supercritical fluid (SCF) blowing agent employed in the microcellular injection molding (MIM) process promotes the oriented dispersion of nanoclays.

### 3.3. Wettability

The wettability of a scaffold plays a critical role in tissue engineering as it significantly impacts cell adhesion. The hydrophilicity of the scaffold surface is known to influence cell proliferation. Therefore, the water contact angle, which reflects the wettability of TPUNC porous scaffolds, was measured at different time intervals to assess their hydrophilicity. Considering the dispersed morphology of the nanoclay filler in the polymer matrix, which can be dispersed in water using ultrasound, it is expected that the addition of nanoclay would enhance the hydrophilicity of TPUNC porous scaffolds by reducing the water contact angle.

Figure 4 presents images of water droplets on the surfaces of TPU, TPUNC5, and TPUNC10 porous scaffolds at different time intervals. Initially, all three porous scaffolds exhibited contact angles greater than 90°, indicating their hydrophobic nature. However, the contact angle of the TPU porous scaffold gradually decreased with time. As depicted in Figure 1, on the first day, the contact angle of the TPU porous scaffold increased with the incorporation of nanoclay compared to the unfilled TPU porous scaffold. The introduction of nanoclay did not initially improve the contact angle of the TPU porous scaffold. Nonetheless, as time progressed, the contact angle gradually decreased with increasing nanoclay content. On the third day, the contact angle (θ) of TPU was 100.4°, while TPUNC5 exhibited a decreased contact angle (θ) of 90.6° with 5% nanoclay, and TPUNC10 showed further reduction to 89.3° with 10% nanoclay. After the tenth day, the contact angle of TPU remained relatively stable compared to the third day, with TPUNC5 exhibiting a decreased contact angle (θ) of 81.7° and TPUNC10 demonstrating a reduced contact angle (θ) of 77.7°. These results indicate that the addition of the SCF foaming agent improves the dispersion of nanoclay in the TPU porous scaffold due to its favorable compatibility with TPU [43].

From Figure 4 and Figure 5, it can be observed that the water contact angle of TPU nanoclay composites is lower than that of TPU, signifying an increase in the hydrophilicity of TPU porous scaffolds with increasing nanoclay content. Based on the observations in Figure 5, it can be observed that the mass of the TPU porous scaffold remained relatively stable after three days of degradation, suggesting a slower degradation rate of the TPU matrix material. However, on the tenth day of degradation, the mass of TPUNC5 and TPUNC10 samples exhibited significant changes, indicating that the porous scaffolds of TPU nanoclay composites underwent faster degradation compared to TPU tissue scaffolds.

### 3.4. Mechanical Properties

Figure 6 presents the results of the relative stress softening, relative hysteresis loss, and relative residual strain of unfilled TPUNR foams under cyclical compression at different maximum strains. In Figure 6a, it is evident that the 40% maximum strain caused a slight stress softening compared to the 20% maximum strain within the same compression cycle. As the maximum compression strain increased to 60%, the relative stress softening became more pronounced compared to that at the 40% strain within the same compression cycle. Furthermore, it is noteworthy that the relative stress softening exhibited a decreasing trend as the compression cycles progressed.

Figure 6b reveals that the relative hysteresis loss exhibits a clear increase as the compression strain increases. Notably, there is a significant disparity in relative hysteresis loss between the first loading-unloading cycle and subsequent cycles, consistent with previous findings in the literature [44]. Subsequent to the first loading-unloading cycle, the relative hysteresis loss becomes less influenced by the number of cycles employed. Moreover, the difference in relative hysteresis loss between the 20% and 40% maximum strains is greater compared to the difference between the 60% and 40% maximum strains. Consequently, while the relative hysteresis loss of the TPUNR foam increases with the maximum compression strain, the rate of increment decreases as the compression strain increases. Figure 6c elucidates that the relative residual strain escalates as the maximum strain intensifies within the same compression cycle. In the case of unfilled, neat TPUNR foams, the relative residual strain becomes more pronounced with higher maximum strains. Concurrently, the relative residual strain exhibits an upward trend with increasing compression cycles. This behavior arises due to the amplified permanent deformation of the cellular structure resulting from higher compression strains and a greater number of compression cycles, consequently leading to an augmented residual strain. A comprehensive analysis of the deformation mechanism of the foams will be presented in detail below.

The aforementioned findings provide evidence that the inelastic behaviors, namely relative stress softening, relative hysteresis loss, and relative residual strain, are notably influenced by the compression conditions, specifically the maximum compression strain. As the compression strain increases, the compressive strength of TPU foams also increases, rising from 0.08 MPa at 20% strain to 0.13 MPa at 40% strain and further to 0.31 MPa at 60% strain, as demonstrated in Figure 6d. Even upon release of the compression stress, a residually deformed structure persists. This permanent deformation contributes to an elevation in hysteresis loss and residual strain. Due to the presence of residual strain from previous compression cycles, a lower compression stress can yield the same maximum compression strain in subsequent cycles, indicating a stress softening behavior. The degrees of relative hysteresis loss, relative residual strain, and relative stress softening all exhibit an increasing trend with the maximum compression strain for TPUNR foam samples. Thus, for TPUNR foams with identical cell structures, their inelastic behaviors become more pronounced at higher maximum compression strains.

Figure 7 illustrates the compressive stress-strain curves of TPUNR, TPUNC5, and TPUNC10 foams subjected to cyclic loading with a recompression strain of 40%. It is evident that the relative stress softening exhibits a significant decrease with increasing nanoclay content, particularly in the case of the TPUNC10 tissue scaffold. Furthermore, for a given compressive strain, the initial cycle compressive strength of the TPU composite enhances with higher nanoclay incorporation. This observation suggests that nanoclay particles exert a notable influence on the inelastic behavior of TPU composites. In addition to the mechanical property enhancement resulting from the nanoclay addition, the presence of oriented nanoclay fillers within the pore walls enhances the gas barrier properties of TPU. During compression of TPU foam, the gas within the closed pores also undergoes compression. Upon pressure release, the compressed gas assists in restoring the deformed structure to its original state. In this study, TPUNC10 foam demonstrated superior gas barrier properties among all the TPU foams due to its elevated nanoclay content and increased number of cell walls per unit volume. The improved gas barrier properties of the foam also contribute to its rebound behavior [19].

### 3.5. Cytocompatibility

To assess the potential application of TPU nanocomposites in tissue engineering, MG63 cells were cultured on selected TPUNC tissue scaffolds for 3 and 10 days. The initial cytocompatibility of microcellular TPUs was assessed by testing cell viability as well as proliferation. To mitigate the complexity and consumption of the ornamentation, TPU, TPUNC5, and TPUNC10 scaffolds were selected as typical representatives of each group. Fluorescent images of live/dead cells on the scaffolds after culturing for 3 and 10 days were obtained and analyzed. Green fluorescence represented live cells, while red fluorescence represented dead cells. The images revealed that very few dead cells (red) were detected, indicating that the cells grew well on the scaffolds during the culture period. Comparing the live/dead cells intuitively, the MG63 cells exhibited similar growth and attachment on the TPUNR and TPUNC scaffolds after 3 days of culture. The cell number appeared to be consistent across the TPUNR, TPUNC5, and TPUNC10 scaffolds, as observed in the images presented in Figure 8. Furthermore, the individual cell morphology exhibited a flattened spindle shape, suggesting favorable cell-material interactions with the incorporated nanofillers. After 10 days of culture, the cells proliferated extensively on the scaffolds, indicating that the TPUNC tissue scaffolds provided a favorable environment for cell attachment and proliferation. From the fluorescence images alone, it was difficult to determine whether the addition of nanoclays to the TPU had a positive or negative effect on cell viability. However, the images demonstrated that the entire scaffolds were covered with cells, and the blurred cell-to-cell boundaries indicated the secretion of a large amount of extracellular matrix by the cells on the scaffold surface. Therefore, both the TPUNR and TPUNC scaffolds exhibited good biocompatibility.

To further investigate the propagation behavior of cells in the TPU scaffolds, the number of MTS-proliferating cells was quantified from MG63 cells on the scaffold at days 3 and 10, as depicted in Figure 9. Comparative analysis of the cell numbers on the TPUNR and TPUNC scaffolds indicated that the addition of nanoclays did not result in a reduction in cell viability. Specifically, there was a slight decrease in cell number from 0.96 × 10^5^ for TPUNR to 0.85 × 10^5^ for TPUNC5 on day 10. However, a subsequent slight increase in cell number from 0.85 × 10^5^ for TPUNC5 to 0.94 × 10^5^ for TPUNC10 was observed, indicating that the TPUNC10 scaffold was also suitable for cell growth. Overall, there was minimal disparity in the number of cells between days 3 and 10, implying that the addition of nanoclay did not significantly impact the cytocompatibility of TPU. In summary, the incorporation of nanoclays preserved the cytocompatibility of TPU and endowed the TPU tissue scaffolds with favorable elastic resilience properties. Therefore, TPU/nanoclay composites exhibit promising potential in the field of tissue engineering.

## 4. Conclusions

The experimental results show that the prepared microcellular TPU/nanoclay composite scaffolds have obvious morphological characteristics and compression properties. The following conclusions are drawn from the experimental results:Scanning electron microscopy (SEM) analysis showed that the surface of the scaffolds exhibited uniform nanoclay dispersion, confirming the effective dispersion of nanoclay in the TPU matrix. In addition, optical microscopy observation showed that the scaffolds presented a delicate microcellular structure with regularly arranged pores, indicating effective foaming control during the preparation process;Compression test results showed that the microcellular TPU/nanoclay composite scaffolds exhibited excellent compression performance. During compression, the scaffolds exhibited linear stress-strain curves and maintained stable mechanical properties when a certain strain was reached. This indicates that the scaffold has a controllable deformation behavior when subjected to compressive loads and has a good energy absorption capacity;Nanoclay was successfully incorporated into TPU using electrostatic spinning with varying nanoclay contents (5 wt% and 10 wt%), resulting in TPUNC tissue scaffolds with excellent interfacial bonding between nanoclay and TPU;Water contact angle tests revealed that all three porous scaffolds initially exhibited hydrophobicity, but with increased time, the contact angle of the TPU porous scaffold gradually decreased as the nanoclay content increased, indicating improved wettability. Moreover, the porous TPUNC tissue scaffold demonstrated superior biocompatibility and biodegradability;Thermogravimetric analysis demonstrated enhanced thermal stability in the TPUNC composites. Tensile tests revealed that the inclusion of nanoclay led to increased tensile modulus and tensile strength in the TPUNC tissue scaffold, with the extent of improvement corresponding to the nanoclay content;Biocompatibility assessment using MG63 cells demonstrated that the TPUNC tissue scaffold exhibited varying effects on cell growth depending on the nanoclay content. Specifically, 5% nanoclay induced excessive oxidative stress in some cells, resulting in cell death. However, increasing the nanoclay content to 10% improved cell adhesion and proliferation on the tissue scaffold, making it a suitable addition for promoting cell growth;Taken together, the results clarify the relationship between the morphological characteristics and compressive properties of microcellular TPU/nanoclay composite scaffolds. Composite scaffolds are potentially valuable as support and cushioning materials in biomedical, engineering, and other applications and provide useful references for further research and applications in related fields.

## Figures and Tables

**Figure 1 polymers-15-03647-f001:**
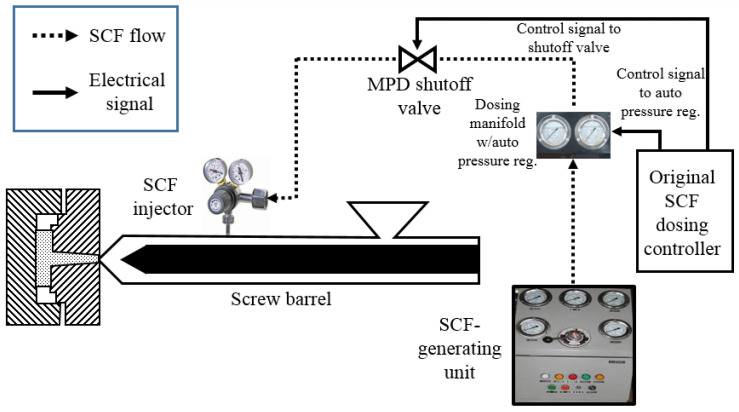
A schematic of the microcellular injection molding process used.

**Figure 2 polymers-15-03647-f002:**
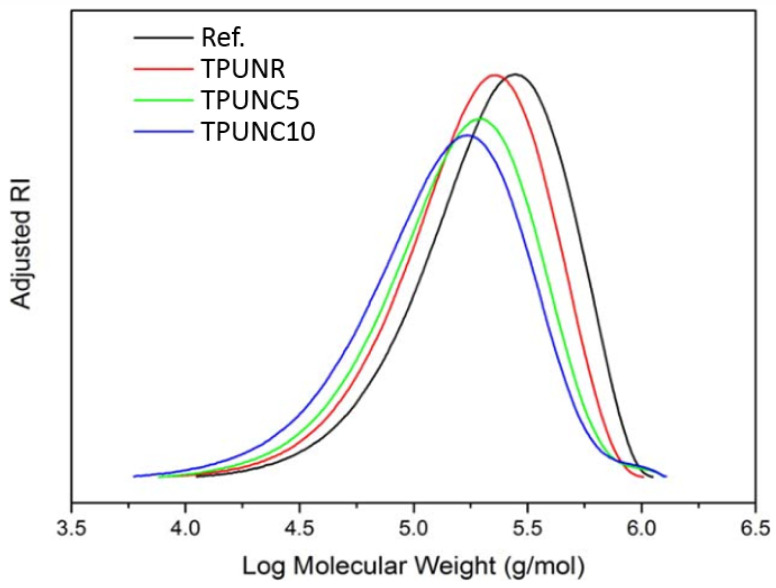
The molecular weight distribution of the reference TPU and the TPUNC system.

**Figure 3 polymers-15-03647-f003:**
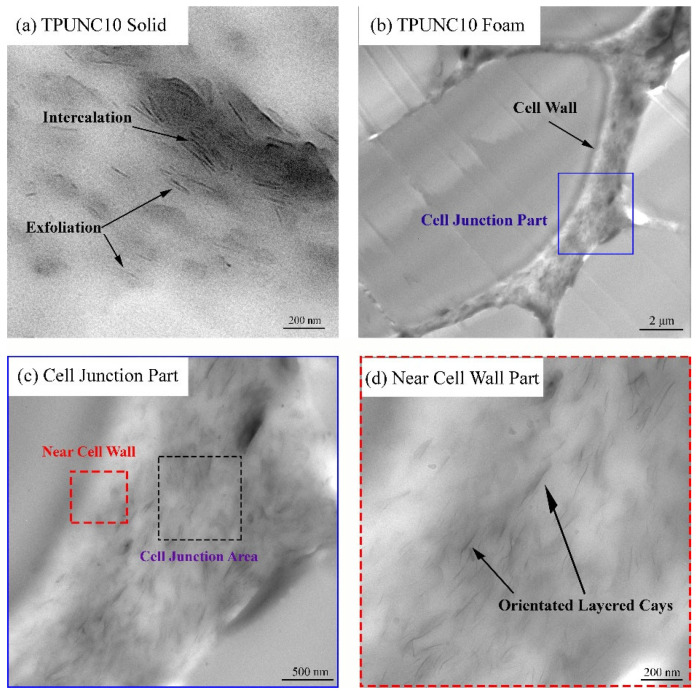
TEM micrographs of the nanoclay morphology from TPUNC10 solid and foam transverse sections.

**Figure 4 polymers-15-03647-f004:**
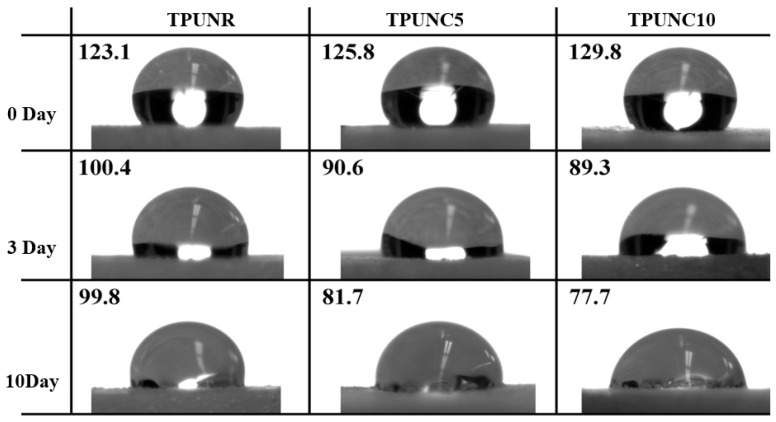
Wettability results of TPU, TPUNC5, and TPUNC10 porous scaffolds with in vitro degradation after 0, 3, and 10 days, respectively.

**Figure 5 polymers-15-03647-f005:**
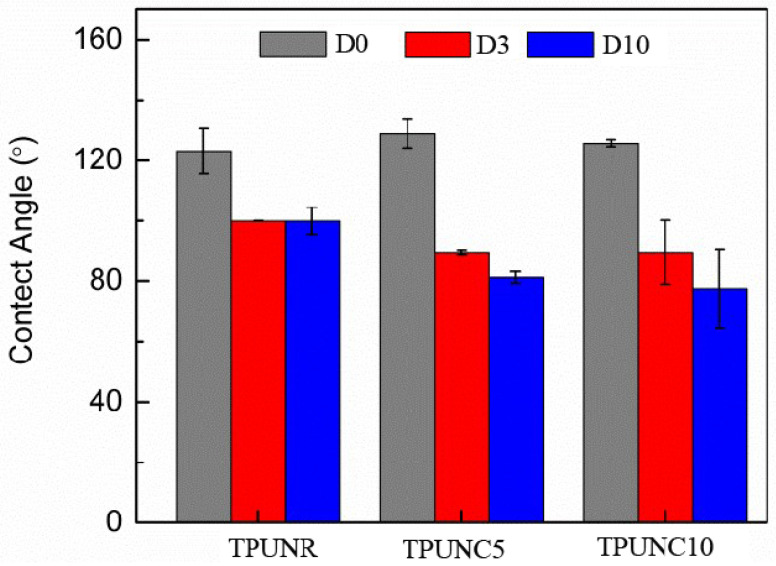
The water contact angles of three TPU scaffolds after degradation for 0, 3, and 10 days.

**Figure 6 polymers-15-03647-f006:**
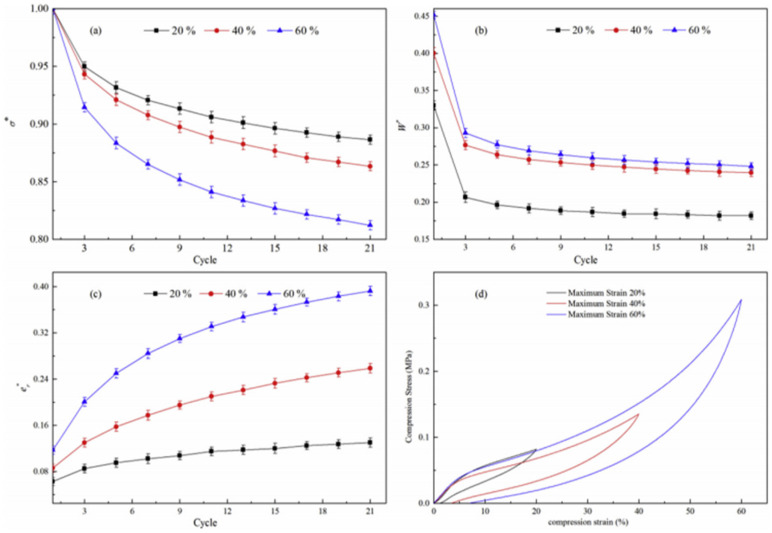
(**a**) The relative stress softening, (**b**) the relative hysteresis loss, (**c**) the relative residual strain, and (**d**) the compression stress–strain curves for the first cycle of TPUNR foams in uniaxial cyclical compression to different maximum strains.

**Figure 7 polymers-15-03647-f007:**
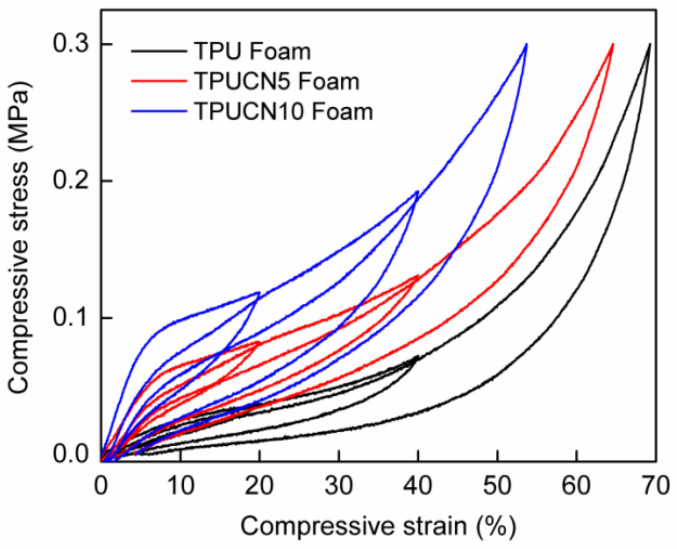
The compressive stress–strain curves of TPU composite foams with the same compression stress.

**Figure 8 polymers-15-03647-f008:**
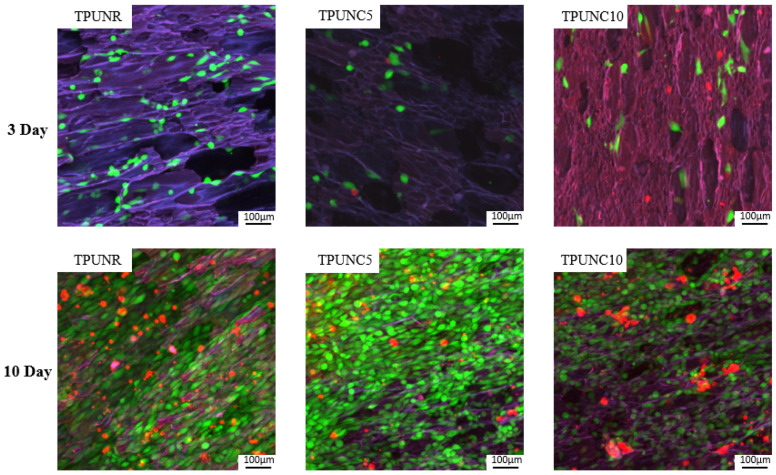
Fluorescence micrographs of stained cells showing live (green) and dead (red) cells on TPUNR, TPUNC5, and TPUNC10 scaffolds after culturing for 3 and 10 days.

**Figure 9 polymers-15-03647-f009:**
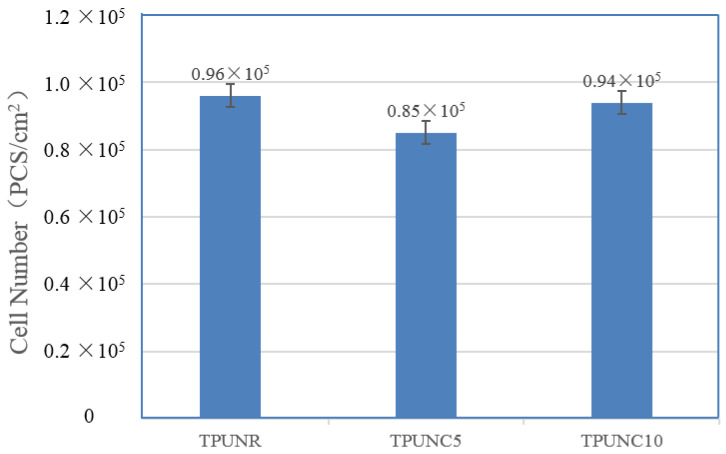
The cell number on the TPUNR, TPUNC5, and TPUNC10 scaffolds after culturing for 10 days.

**Table 1 polymers-15-03647-t001:** Experimental processing parameters for microcellular injection molding.

Parameter	Value
Injection speed (cm^3^/s)	50
Mold temperature (°C)	45
Cooling time (s)	13
SCF flow rate (kg/h)	0.8
SCF injection pressure (MPa)	30

## Data Availability

Not applicable.

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
