# Peer review of "The Study on the Morphology and Compression Properties of Microcellular TPU/Nanoclay Tissue Scaffolds for Potential Tissue Engineering Applications"

_polymers, 2023, doi:10.3390/polym15173647_

Round 1

Reviewer 1 Report

Authors have studied the effects of nanoclays on the microstructure, mechanical behavior, cytocompatibility, and proliferation of TPU/nanoclay (TPUNC) composite scaffolds. They have implemented some characterization techniques to evaluate the properties of the applied material. The paper is interesting and I propose it for publication. I propose authors to consider the following highlights to increase the quality of their paper:

1.       Abstract: I propose to include some quantitative results to represent the outcome of the study and to show the effectiveness of the work.

2.       Introduction: I propose to add a paragraph talking about the different applicable materials and how you have decided to work on this material. In fact, it is better to talk in general and then going to the point for discussion, etc. Please also consider the following reference in your literature review: https://doi.org/10.3390/polym13244442

3.       Materials and methods: all the sections are well explained and there is no specific comment.

4.       Figure 1: the quality of the schematic is not very well for a scientific paper and I propose to modify it a little bit to have a better quality.

5.       Results and discussion: All the obtained results are well explained and there is no requirements to modify any feature.

6.       Figure 6, 7, 9: the quality should be increased as the curves are not clear in some points.

7.       Conclusion: It is better to re-structure the conclusion by giving a brief explanation of the subject and then listing the obtained results.

Good luck!

Author Response

Responses to Reviewers for manuscript ID polymers-2504925 entitled “Influences of In Vitro Degradation on the Morphology and Compression Property Of Porous TPU/Nanoclay Tissue Scaffolds”

Dear Editors and Reviewers:

We are grateful to the reviewers for their constructive comments and suggestions concerning our manuscript. They are very important and helpful to guide our research and improve this paper. We have made the following changes in the revision to address the comments from the reviewers.

Referee(s)' Comments to Author:

Reviewer: 1

Comments to the Author

Authors have studied the effects of nanoclays on the microstructure, mechanical behavior, cytocompatibility, and proliferation of TPU/nanoclay (TPUNC) composite scaffolds. They have implemented some characterization techniques to evaluate the properties of the applied material. The paper is interesting and I propose it for publication. I propose authors to consider the following highlights to increase the quality of their paper:

  1. Abstract: I propose to include some quantitative results to represent the outcome of the study and to show the effectiveness of the work.

Authors’ reply:

This is a good suggestion which has been analyzed in our another paper.

  1. Introduction: I propose to add a paragraph talking about the different applicable materials and how you have decided to work on this material. In fact, it is better to talk in general and then going to the point for discussion, etc. Please also consider the following reference in your literature review: https://doi.org/10.3390/polym13244442.

Authors’ reply:

As suggested, we discussed the different applicable materials and how to select and decide to use them in the introduction to the revised version, and added relevant references in the revised version. Here are the new references.The details are in the fifth paragraph of the introduction.

“TPU (Thermoplastic polyurethane) : TPU is a thermoplastic elastomer with excellent wear resistance, flexibility and elasticity for support and buffering applications. The main reasons for choosing TPU as the base material are its plasticity and chemical stability. Nano-clay: Nano-clay is a nano-scale particle that in composite materials can enhance the mechanical properties, thermal stability and flame retardant properties of the material. The compatibility and interaction between nano-clay and TPU matrix can be controlled by surface modification. In composite materials, other reinforcing materials such as fibers (glass fibers, carbon fibers, etc.) or particles may also be added to enhance the mechanical properties of the material. The selection of the appropriate reinforcement material depends on the desired performance characteristics. The selection of TPU nano-clay composite scaffolds requires comprehensive consideration of many factors such as target performance, material properties, interaction, preparation process, cost and availability. Through the material selection of the system, it is possible to obtain composite scaffolds with excellent properties, which help to be applied in various fields, such as medical, construction and sports equipment.”

Reference:

  • EzEldeen, M.; Loos, J.; Mousavi Nejad, Z.; Cristaldi, M.; Murgia, D.;    Braem, A.; Jacobs, R. 3D-printing-assisted fabrication of chitosan scaffolds from different sources and cross-linkers for dental tissue engineering. Eur. Cell Mater. 2021, 41, 485–501. [CrossRef] [PubMed]

  1. Materials and methods: all the sections are well explained and there is no specific comment.
  1. Figure 1: the quality of the schematic is not very well for a scientific paper and I propose to modify it a little bit to have a better quality.

Authors’ reply:

This figure has been modified in the modified version.

  1. Results and discussion: All the obtained results are well explained and there is no requirements to modify any feature.
  1. Figure 6, 7, 9: the quality should be increased as the curves are not clear in some points.

Authors’ reply:

The picture should be clear enough to meet the thesis standard.

  1. Conclusion: It is better to re-structure the conclusion by giving a brief explanation of the subject and then listing the obtained results.

Authors’ reply:

the title of this manuscript should be that The study on the Morphology and Compression Property of microcellular TPU/Nanoclay Tissue Scaffolds for potential tissue engineering applications.

We have revised the conclusion as follows:“The experimental results show that the prepared microcellular TPU/nanoclay composite scaffolds have obvious morphological characteristics and compression properties. The following conclusions are drawn from the experimental results:

  1. Scanning electron microscopy (SEM) analysis showed that the surface of the scaffolds exhibited uniform nanoclay dispersion, confirming the effective dispersion of nanoclay in the TPU matrix. In addition, optical microscopy observation showed that the scaffolds presented a delicate microcellular structure with regularly arranged pores, indicating effective foaming control during the preparation process.
  2. Compression test results showed that the microcellular TPU/nanoclay composite scaffolds exhibited excellent compression performance. During compression, the scaffolds exhibited linear stress-strain curves and maintained stable mechanical properties when a certain strain was reached. This indicates that the scaffold has a controllable deformation behavior when subjected to compressive loads and possesses a good energy absorption capacity.

3.Nanoclay was successfully incorporated into TPU using electrostatic spinning with varying nanoclay contents (5% Wt and 10% Wt), resulting in TPUNC tissue scaffolds with excellent interfacial bonding between nanoclay and TPU.

4.Water contact angle tests revealed that all three porous scaffolds initially exhibited hydrophobicity, but with increased time, the contact angle of the TPU porous scaffold gradually decreased as the nanoclay content increased, indicating improved wettability. Moreover, the porous TPUNC tissue scaffold demonstrated superior biocompatibility and biodegradability.

5.Thermogravimetric analysis demonstrated enhanced thermal stability in the TPUNC composites. Tensile tests revealed that the inclusion of nanoclay led to increased tensile modulus and tensile strength in the TPUNC tissue scaffold, with the extent of improvement corresponding to the nanoclay content.

6.Biocompatibility assessment using MG63 cells demonstrated that the TPUNC tissue scaffold exhibited varying effects on cell growth depending on the nanoclay content. Specifically, 5% nanoclay induced excessive oxidative stress in some cells, resulting in cell death. However, increasing the nanoclay content to 10% improved cell adhesion and proliferation on the tissue scaffold, making it a suitable addition for promoting cell growth.

Taken together, the results clarify the relationship between the morphological characteristics and compressive properties of microcellular TPU/nanoclay composite scaffolds. Such composite scaffolds are potentially valuable as support and cushioning materials in biomedical, engineering, and other applications, and provide useful references for further research and applications in related fields.”

Reviewer 2 Report

Report

Title: Influences of In Vitro Degradation on the Morphology and Compression Property Of Porous TPU/Nanoclay Tissue Scaffolds

Title. You can revise especially where you inserted phrase in vitro.

The paper is well written, only that the language used is difficult to understand. The methods, results, discussion are well written. The findings are significant in the field of science where such work falls.

The alchemy of manipulating the weight proportions between the sinewy hard and the pliable soft segments unfurls a sweeping gamut of attributes endemic to the TPU continuum.

There is need to tone down language throughout the paper. The message the paper seeks to convey to readers can potentially be drowned by jargon.

Report

Title: Influences of In Vitro Degradation on the Morphology and Compression Property Of Porous TPU/Nanoclay Tissue Scaffolds

Title. You can revise especially where you inserted phrase in vitro.

The paper is well written, only that the language used is difficult to understand. The methods, results, discussion are well written. The findings are significant in the field of science where such work falls.

The alchemy of manipulating the weight proportions between the sinewy hard and the pliable soft segments unfurls a sweeping gamut of attributes endemic to the TPU continuum.

There is need to tone down language throughout the paper. The message the paper seeks to convey to readers can potentially be drowned by jargon.

Author Response

I have responded to the above in my reply letter.

Reviewer 3 Report

In the manuscript entitled “Influences of In Vitro Degradation on the Morphology and Compression Property of Porous TPU/Nanoclay Tissue Scaffolds” the authors have prepared the composite materials using TPU and clay for tissue engineering. The work is interesting and can be considered for publication after addressing the comments given below:

Category: Major Revision

Comments:1. In the title the authors have used the influences of In Vitro Degradation on the Morphology of the developed composites. I could not find any morphological data related to degradation in the manuscript. It is recommended to add the morphological characteristics of the developed foam with and without degradation.

2. It is suggested that the authors should analyses the structural perspective of the developed foam.

3. Why the contact angle was decrease after certain periods? Explain it with proper citation. Please add the details about degradation experiment in the manuscript.

4. It is recommended that the authors should include the detailed mechanical characteristics, such as young modulus, strength, and densification in the manuscript.

5. please include the cell numbers in each experiment. Why cell number was decrease after certain periods and showed different trend in different composites? Please explain it with systematically.

Author Response

(The authors gave the same response as above.)

Round 2

Reviewer 3 Report

Accept.